# The Brain Protein Acylation System Responds to Seizures in the Rat Model of PTZ-Induced Epilepsy

**DOI:** 10.3390/ijms232012302

**Published:** 2022-10-14

**Authors:** Lev G. Zavileyskiy, Vasily A. Aleshin, Thilo Kaehne, Irina S. Karlina, Artem V. Artiukhov, Maria V. Maslova, Anastasia V. Graf, Victoria I. Bunik

**Affiliations:** 1Faculty of Bioengineering and Bioinformatics, Lomonosov Moscow State University, 119234 Moscow, Russia; 2Department of Biokinetics, A.N. Belozersky Institute of Physicochemical Biology, Lomonosov Moscow State University, 119234 Moscow, Russia; 3Department of Biochemistry, Sechenov University, 119048 Moscow, Russia; 4Institute of Experimental Internal Medicine, Otto von Guericke University, 39106 Magdeburg, Germany; 5N.V. Sklifosovsky Institute of Clinical Medicine, Sechenov First Moscow State Medical University, 119435 Moscow, Russia; 6Faculty of Biology, Lomonosov Moscow State University, 119234 Moscow, Russia

**Keywords:** acylation of brain protein, PTZ-model of epilepsy, brain energy metabolism, SIRT2, SIRT3, SIRT5, 2-oxo acid dehydrogenase

## Abstract

Abnormal energy expenditure during seizures and metabolic regulation through post-translational protein acylation suggest acylation as a therapeutic target in epilepsy. Our goal is to characterize an interplay between the brain acylation system components and their changes after seizures. In a rat model of pentylenetetrazole (PTZ)-induced epilepsy, we quantify 43 acylations in 29 cerebral cortex proteins; levels of NAD^+^; expression of NAD^+^-dependent deacylases (SIRT2, SIRT3, SIRT5); activities of the acyl-CoA-producing/NAD^+^-utilizing complexes of 2-oxoacid dehydrogenases. Compared to the control group, acylations of 14 sites in 11 proteins are found to differ significantly after seizures, with six of the proteins involved in glycolysis and energy metabolism. Comparing the single and chronic seizures does not reveal significant differences in the acylations, pyruvate dehydrogenase activity, SIRT2 expression or NAD^+^. On the contrary, expression of SIRT3, SIRT5 and activity of 2-oxoglutarate dehydrogenase (OGDH) decrease in chronic seizures vs. a single seizure. Negative correlations between the protein succinylation/glutarylation and SIRT5 expression, and positive correlations between the protein acetylation and SIRT2 expression are shown. Our findings unravel involvement of SIRT5 and OGDH in metabolic adaptation to seizures through protein acylation, consistent with the known neuroprotective role of SIRT5 and contribution of OGDH to the Glu/GABA balance perturbed in epilepsy.

## 1. Introduction

Abnormal activity of the brain, which results in a disbalance between neuronal excitation and inhibition, may cause seizures. The latter is a hallmark of epilepsy—a common disease with a predominantly polygenic origin, characterized by perturbed interplay between the glutamate and GABA neurotranstmission [1,2,3,4,5]. During seizures, cerebral blood flow and metabolic rate of glucose and oxygen consumption increase to address enormous energy demands [6]. Nevertheless, the energy shortage due to limitations in oxygen supply enhances the relative energetic contribution of glycolytic pathway vs. oxidative phosphorylation [7]. A preventive effect of oxygen therapy in generalized convulsive seizures supports hypoxemia as an immediate result of the seizures [8]. Thus, seizures induce significant changes in the brain energy metabolism.

Regulation of proteins by post-translational modifications (PTM) participates in adaptive reprogramming of metabolic fluxes. While epigenetic mechanisms involving acetylation of histones are long-known, lately also other types of acylation, and acylation of metabolic proteins have drawn increasing attention regarding metabolic regulation [9,10,11,12,13,14,15,16,17,18].

Pyruvate dehydrogenase (PDH) is an important enzyme linking glycolysis to oxidative phosphorylation, which is tightly regulated, also by PTM. The multienzyme complex of PDH produces acetyl-CoA—the substrate for protein acetylation. Other members of the family of the 2-oxo acid dehydrogenase complexes, namely the complexes of 2-oxoglutarate dehydrogenase (OGDH) and 2-oxoadipate dehydrogenase (OADH), are important producers of the negatively charged acyl-CoA, i.e., succinyl- and glutaryl-CoA respectively, for the corresponding protein acylations [19,20]. Thus, the acyl-CoA products of the 2-oxoacid dehydrogenase complexes may serve as donors of acyl residues in acyl-transferase reactions involving the protein lysine residues. This view is independently supported by our studies that reveal association between the protein acetylation and the actions of thiamine (vitamin B1) [15,17,21], whose diphosphorylated form, thiamine diphosphate, is a coenzyme required for the catalytic function of the 2-oxoacid dehydrogenase complexes.

On the other hand, metabolic proteins are deacylated by the acyl-specific NAD^+^-dependent deacylases, i.e., sirtuins. Sirtuin 2 (SIRT2) is the major cerebral deacetylase in cytosol, sirtuin 3 (SIRT3) is the mitochondrial deacetylase and sirtuin 5 (SIRT5), localized both to mitochondria and cytosol, is specific to negatively charged acylations, i.e., malonylation, succynilation and glutarylation [22,23,24,25].

We hypothesize that significant metabolic and signaling disbalances in the epilepsy-affected brain are manifested in the brain protein acylation system, whose regulation may represent a therapeutic target. The goal of this work is to characterize the brain protein acylation system in a rat model of epilepsy. Among the components of the brain protein acylation system, we consider (i) the central enzyme complexes of mitochondrial energy metabolism, that simultaneously are the acyl-CoA producers and NAD^+^ consumers, i.e., the complexes of PDH and OGDH, together with the complex of the DHTKD1-encoded OADH producing glutaryl-CoA; (ii) the brain-abundant NAD^+^-dependent deacylases, i.e., SIRT2, SIRT3, SIRT5; and (iii) the levels of NAD^+^. In view of the involvement of vitamin B1 with acetylation of metabolic proteins [17,21], and our previous data on the protective effects of the combined administration of vitamins B1 and B6 in the widely used model of pentylenetetrazole (PTZ)-induced seizures [26], these metabolic regulators are included in the current study as factors potentially affecting the brain acylation system under physiological and/or pathological states. Besides, taking into account the brain damage accumulation with chronic exposure to seizures [27,28], we employ a comparison of the PTZ-induced single seizures [26] and chronic seizures upon pharmacological kindling with PTZ [29]. We show how the brain protein acylation system is altered in response to the single and chronic seizures, and which mechanisms are the most relevant for this process.

## 2. Results

### 2.1. Seizures-Induced Alteration in the Brain Protein Acylation Levels Involves Enzymes of Central Energy Metabolism, with More Acylation Sites Responding to Single Than Chronic Seizures

To characterize changes in the brain acylation system due to seizures, we use an established animal model of the epileptic seizures (see Section 4.2 under Materials and Methods) induced by PTZ, the binding of which to GABA_A_ receptors perturbs the balance of the excitatory and inhibitory signals, known to be a hallmark of epilepsy [30]. The brain protein acylation levels identified by LC-MS/MS in the rat brain cortex homogenates of the control animals and those after a single seizure episode or recurrent seizures are compared in order to reveal the seizure consequences for the post-translational modifications, and the dependence of these consequences on the seizure recurrence.

In total, 99 acylation sites in 62 proteins have been identified by our LC-MS/MS approach in the rat brain cortex homogenates. However, only 43 sites in 29 proteins had a signal-to-noise ratio sufficient for quantification in all the studied groups (Appendix A). From those, 14 acylation sites significantly change after PTZ-induced seizures (Figure 1). Among the quantified sites, administration of the vitamins B1 and B6 significantly affects glutarylation of CN37 K234, while other acylation levels do not exhibit a significant response to the vitamins (Figure 1, Appendix A). More sites respond to single seizures (12 of 14), compared to chronic seizures (9 of 14), suggesting that a strong perturbation of the brain acylation system by a seizure is followed by adaptive response due to the chronic condition. However, the difference between the effects of the single and chronic seizures does not reach statistical significance. A majority of the sites demonstrate a similar pattern to the seizure-induced changes: Compared to the control state, seizures significantly lower the acylation level in one or both groups. Yet other responses are not excluded: glutarylation of BASP1 after both seizure types does not decrease but increases.

More than a half of the sites (8 of 14), whose level of acylation is altered by seizures (Figure 1), belong to the enzymes involved in central energy metabolism. Those include the glycolytic enzymes hexokinase (HXK1) and phosphoglycerate mutase (PGAM1), glycolysis-affiliated lactate dehydrogenase (LDHA), as well as mitochondrial proteins, such as the second component of PDH complex (ODP2); acetyl-CoA acetyltransferase (THIL) participating in the fatty acid beta-oxidation; the ADP/ATP antiporter (ADT2) that mediates the import of ADP for intramitochondrial ATP synthesis and efflux of ATP to fuel the cell energy demands. Thus, the abnormal energy expenditure in the PTZ-induced seizures is associated with alterations in the acylation level of the proteins involved in central energy metabolism. Of these proteins, acetylation and malonylation of lactate dehydrogenase, as well as glutarylation of acetyl-CoA acetyltransferase demonstrate adaptive responses to chronic seizures, which significantly decrease vs. control animals only after the single seizure episode. In contrast, the changes in the glutarylation of hexokinase and the second component of PDH complex vs. control animals are significant only after chronic seizures. Finally, the changes in acylations of phosphoglycerate mutase and ADP/ATP antiporter, observed after a single seizure, persist under chronic seizures.

### 2.2. Activity of OGDH Complex and Expression of SIRT3 and SIRT5 Are Lower in Chronic vs. Single Seizures, While Activity of PDH Complex and Expression of SIRT2 Remain Unchanged

Perturbation in the brain protein acylation system, underlying the seizures-induced changes in the protein acylation (Figure 1), could involve the functional state of the multienzyme complexes of 2-oxo acid dehydrogenases (PDH, OGDH and OADH), NAD^+^ levels and expression of SIRT2, SIRT3, SIRT5. Quantifications of these parameters in the studied brain cortex samples are shown in Figure 2.

Activity of a PDH complex and expression of SIRT2, representing the components of the protein acetylation system, as well as NAD^+^ content, remain at the same level in all the groups (Figure 2A–C). In contrast, the mitochondrial deacetylase SIRT3 and contributors to the negatively charged acylations (OGDH, SIRT5) are significantly lower in chronic seizures compared to single seizures (Figure 2A,C). Besides, compared to the control rats, SIRT3 is also lower after chronic seizures, while SIRT5 increases after a single seizure (Figure 2C). Remarkably, administration of the vitamins B1 and B6 increases the levels of NAD^+^, as judged by the statistical significance of the treatment revealed by ANOVA analysis of all the groups (*p* < 0.01, Figure 2B).

The simultaneous decrease in the OGDH function and SIRT5 expression under chronic vs. single seizures (Figure 2A,C) is supported by a significant positive correlation between OGDH and SIRT5 (Spearman r = 0.468, *p* = 0.014). Remarkably, the decreased function/expression of the contributors to the negatively charged acylation (OGDH, SIRT5) after the chronic vs. single seizures is accompanied by no significant differences in the acylation levels between the single and chronic seizures (Figure 1, Appendix A). The relationship demonstrates the OGDH-supported compensatory mechanism alleviating the seizure-induced perturbations in the brain protein acylation due to an increase in SIRT5 (after a single seizure), or a decrease in SIRT3 (after chronic seizures).

### 2.3. Correlations between Different Types of Protein Acylation and acyl-CoA Producers, Acyl-Specific Sirtuins and NAD^+^

To further characterize the interplay between the protein acylation levels and the studied components of the brain acylation system, pairwise correlations of the parameters values across all the animal groups have been assessed (Figure 3).

Regarding the studied acyl-CoA producers, multiple significant positive correlations between the glutarylation level of a number of proteins and expression of the glutaryl-CoA producer OADH are revealed, testifying to an essential role of OADH expression in glutarylation of the brain proteins. In contrast, the correlations between the protein acetylation or succinylation levels with the function of the corresponding complexes of PDH or OGDH are not so straightforward. This observation may be due to extensive homeostatic networking of these systems of central energy metabolism, in contrast to the glutaryl-CoA producer OADH.

Significantly more extensive networking of the components of the brain acetylation system vs. those involved in the negative acylation is further supported by the correlations between the protein acylation levels and the acyl-specific deacylases. Levels of the negatively acylated lysine residues are mostly negatively correlated with the expression of SIRT5. In contrast, the levels of all types of acylation, including acetylation, mostly show positive correlations with SIRT2. Only a minor part of the acetylation sites, represented by acetylation of CN37, demonstrates the expected negative correlation of the acetylation level with the expression of this deacetylase. No major trends of the protein acylation level with the expression of mitochondrial deacetylase SIRT3 or total NAD^+^ level are revealed across all the animal groups studied (Figure 3).

It is worth noting that the patterns of correlations with the acyl-CoA producers and sirtuins, revealed across all the groups (Figure 3), are generally preserved when the specific groups are analyzed (Appendix A). Yet, the specific contribution of each group to the pooled correlation may differ, dependent on the parameter and group. For instance, the negative correlations of the negatively charged acylations with SIRT5 are mostly expressed in the control and single-seizure group, the positive correlations of glutarylation level with OADH expression are the strongest after the single seizure, while positive correlations of all acyl types with SIRT2 are present in all the groups (Appendix A).

Overall, the results of our correlation analysis are consistent with significantly more complex homeostatic networks determining the levels of acetylation and succinylation than those of glutarylation.

As protein sites modified by different acyl groups often overlap, potential competition between different acylation reactions for a single lysine residue is analyzed. Three such sites, identified in the studied rat brain homogenates, are shown in Figure 4. The existence of the competitive acylation reactions is confirmed by the negative correlation between the levels of the CN37 acetylation or glutarylation at K177 (Figure 4, CN37). However, in the case of K340 of neurofilament protein, no competition is observed between the residue acetylation or glutarylation (Figure 4, NFL). Finally, malonylation and glutarylation of K451 of ODP2 are positively correlated (Figure 4, ODP2). Thus, competitive acylation of a single protein site is not the only possible option, but the competition between the overlapping acylations may contribute to the complexity of the acylation regulation.

### 2.4. Dependent on the Acylation Site/Protein, Both Negative and Positive Correlations May Be Observed with Acylation of Other Sites, Even for the Same Type of Acylation

The correlation matrix of all acylations (Figure 5, All acylations) reveals a cluster of strong positive correlations between acylations regardless of the acylation type (cluster No. 1). However, this does not exclude negative correlations between other pairs of the acylated lysine residues (cluster No. 3). Even when we consider different sites of acetylation only, the levels of acetylation may show significant negative correlations (Figure 5, Acetylation). Thus, in the studied sample of animals, acetylation of some sites may either increase (positively correlated acetylations) or decrease (negatively correlated acetylation) along with acetylation of other sites. This heterogeneity excludes a view that all acylations of the brain proteins respond uniformly to some metabolic steady state characterized by an acetylation potential of the system.

## 3. Discussion

### 3.1. Altered Acylation System of the Brain after Seizures

In this work, alterations in the levels of acylation of the rat brain proteins and the enzymes contributing to the regulation of these post-translational modifications are identified after single and chronic seizures induced by PTZ. It is known that PTZ induces epilepsy by binding to GABA_A_ receptors [30]. Thus, the changed acylation of the receptor may be a direct consequence of the PTZ action. However, the changed acylation of other proteins identified in this work, manifests complex secondary events of the PTZ binding to GABA_A_ and induces seizures due to a disbalance between the neuronal excitation and inhibition. To the best of our knowledge, our data on specific changes at the levels of acylations of different protein sites after seizures are unique. However, regarding the other components of the brain acylation system studied by us, published data reveal remarkable coincidence of the seizure-induced changes, pointing to their independence from the specific chemical inductor of the seizures. As shown in Table 1, apart from our study, the sirtuin 3 decrease after chronic seizures is known from two other independent studies employing PTZ or kainate, with the decrease also observed in juvenile rats after the single seizure episode induced by pilocarpine. After a single seizure episode, sirtuin 5 increases in our PTZ model of epilepsy, with similar effect observed in the kainate model (Table 1). The hippocampal activities of PDHC and OGDHC decrease after the pilocarpine- and PTZ-induced chronic seizures compared to the control groups (Table 1). In our study of the brain cortex, there was a trend to decreased OGDH activity (*p* = 0.1, chronic seizures compared to control, Figure 2A). Apart from manifesting the tissue-specific effects of the changes in PDHC and OGDH in the hippocampus and cerebral cortex, different activity assays are employed in the studies, potentially contributing to the observed differences in the effects. Nevertheless, the published data confirm the involvement of such components of the acylation system as PDHC and OGDH into the brain response to recurrent seizures (Table 1).

As a result, changes in the brain acylation system after seizures, which we observe in the PTZ model of epilepsy, are supported by independent studies on the models employing varied chemical inductors. Along with the known molecular mechanism of the PTZ action, the accumulated data indicate that the system of the brain acylation change after the chemically induced seizures.

### 3.2. Regulation of Acylation in the Rat Model of PTZ-Induced Epilepsy

According to previous studies, acylations are regulated by acyl-CoA levels and activities of the NAD^+^-dependent deacylases [36,37,38]. Data obtained in this work indicate that many significant changes in acylations upon seizures are unidirectional (Figure 1). These changes correspond to a view on acylation as a chemistry-driven protein response to carbon stress, attenuated by sirtuins [38]. Yet acylations of different protein sites may show not only positive, but also negative correlations between each other (Figure 5). Besides, an increase in glutarylation of BASP1 K158 after seizures vs. controls is seen when most of other significantly changed acylations decrease (Figure 1). Finally, we observe several proteins with positive relationships between their acetylation levels and the expression of major brain NAD^+^-dependent deacetylase SIRT2 (Figure 3). All together, these observations indicate that a more complex view on the regulation of protein acetylation, going beyond a uniform chemical response to carbon stress, is required. As different acylation types may share their sites [24], competitive acylations exemplified by acetylation and glutarylation of CN37 K177 (Figure 3 and Figure 4) may contribute to mutually affected acylation levels. Heterogeneity of the intracellular environment and metabolism in different brain cells may cause local changes, different from the bulk ones, due to the action of the less abundant and site-specific deacylases, and the location-specific metabolism of acyl-CoAs. Apart from the specific spatial organization of biological systems, the different lifetime of the proteins may further contribute to the complexity. In particular, BASP1 has peptide sequences rich in proline, glutamate, serine, and threonine (PEST) [39,40]. Such PEST sequences are markers of the rapidly proteolyzed proteins [41]. Compared to an average half-life of brain proteins estimated to be about 5 days [42], that of BASP1 may be within hours. Thus, the difference between the acylation of BASP1 and other proteins showing significant changes in their acylation levels (Figure 1) may be due to the different time scales of the acylation-affecting response to seizures: Acylation of long-lived proteins memorizes the state of the brain acylation system for a longer time than the acylation of rapidly degraded and re-synthesized BASP1 does. For instance, decreased glutarylation level of the long-lived ADT2 vs. control may manifest an earlier event after the seizures than the increased glutarylation of the short-lived BASP1 vs. control.

As a result, assayed at a fixed point after seizures, acylation of proteins with different lifetimes may mark evolution of the acylation system state after the seizures, as well as spatial differences in the acylation networks.

Apart for the factors mentioned above, extensive homeostatic networking of OGDH and PDH complexes may result in complexity of their correlations with acylations of different type, not observed for the glutaryl-CoA producer OADH (Figure 3). In particular, both PDH and OGDH complexes interact through the tricarboxylic acid (TCA) cycle, where the PDH complex feeds the cycle substrate acetyl-CoA, while OGDH complex limits the substrate flux. This metabolic interaction may manifest in indirect influences on PDH and OGDH on the corresponding protein acylations. For instance, a significant positive correlation of acetylation of 2’,3’-cyclic-nucleotide 3’-phosphodiesterase (CN37) with the OGDH activity may be due to the TCA-cycle-limiting function of OGDH complex, that may be essential for the level of acetyl-CoA directed at the acetylation of CN37. Significant negative correlations of the PDH activity and level of succinylation of mitochondrial malate dehydrogenase (MDHM), or glutarylation of a Ca^2+^ sensor synaptotagmin 1 (SYT1) (Figure 3), suggest a competition between each of these negative acylations with acetylation. Indeed, our data on CN37 confirm the possibility of such a competition (Figure 4). The positive correlation of NFL glutarylation with OGDH activity may be due to the common second and third components of the OGDH and OADH complexes, and the potential contribution of OADH to the assayed mitochondrial activity of OGDH [43].

### 3.3. Involvement of Perturbed Acylation in Neurological Disorders

Our work reveals acylations of several proteins related to epilepsies (GNAO1 [44], SYN1 [45]) and a broad spectra of neurodegenerative diseases (TAU [46,47] and others) (Appendix A). Although we could not detect significantly changed levels in these acylations by seizures, the change is revealed in glutarylation of NFL and HXK1 proteins whose mutations cause different types of Charcot-Marie-Tooth disease [48,49]. Remarkably, this disease is also known to be associated with mutations in the DHTKD1-encoded OADH [50,51], whose expression is strongly correlated with glutarylation of multiple brain proteins (Figure 3). Mutations in ODP2 gene, coding the second component of PDH complex, are accompanied by neurological problems, including seizures [52]. In view of the PDH complex regulation by acylation [12,13], clinical manifestations of PDH mutations outside the active and binding sites, may depend on perturbed acylation of lysine residues of the PDH complex. Significance of such impairments is exemplified by the, the TAU protein whose perturbed acetylation can trigger TAU aggregation through its hyperphosphorylation, that is found in AD, corticobasal degeneration, progressive supranuclear palsy and is that TAU transgenic mice models of tauopathies [53,54]. Succynilation of TAU becomes detectable in AD human brain samples, unlike healthy ones, which may also contribute to AD pathology as shown by increased aggregation capacity of succynilated TAU peptides in vitro [55].

Interestingly, administration of vitamins B1 and B6 increases the level of the deacylases substrate NAD^+^. The finding may be useful for the treatment of the neurodegenerative conditions, which are improved by increasing NAD^+^ levels [56]. It may also explain positive action of vitamins supplementation in epilepsy, observed by us earlier [26]. In fact, another sirtuin activator, resveratrol, is considered as an anticonvulsant agent [57].

Thus, targeting posttranslational acylation might be a general strategy to combat neurological diseases.

### 3.4. Regulation of Energy Metabolism by Acylation upon Seizures

A subcluster of the most correlated acylations revealed in our work (Figure 5, cluster No. 2) is enriched with proteins involved in energy metabolism both in mitochondria (AT1B1, ADT2, ODP2) and cytosol (HXK1, PGAM1, LDHA). The glycolytic enzymes are known to be regulated by acetylation [58]. This is in good accordance with the regulatory function of acylations in epilepsy, as the pathology is known to cause metabolic changes, primarily related to energy metabolism, regardless of the disease origin.

Our and others’ finding of decreased level of SIRT3 after chronic seizures (Table 1) provides further strong support for the significance of the brain energy metabolism in epileptic seizures, as many mitochondrial enzymes involved in energy production are acylated. Hyperacetylation and inactivation of the mitochondrial antioxidant enzyme MnSOD via decreased activity of SIRT3 may prolong epileptic seizures and increase mitochondrial levels of ROS [14,59]. While the roles of mitochondrial acyl transferases are under debate [23,38,58], in cytosol and the nucleus, enzymatic acylation prevails and regulates almost all glycolytic enzymes [58]. Although GLUT1 and HK1 are downregulated by acetylation, inhibiting the start of glycolysis [60], acetylation is known to positively regulate PFKFB3, GAPDH, PGK1, PGAM1, PKM2 [10,61,62,63,64]. Of those, decreased glutarylation of two lysine residues of PGAM1 is shown in our work (Figure 1). The inhibition of the mitochondrial PDH complex by acetylation and succinylation also contributes to activation of glycolysis [12,13]. Acetylation upregulates malate–aspartate shuttle by increasing enzymatic activity of cytoplasmic MDH1 and mitochondrial GOT2, enhancing glycolysis due to stimulation of mitochondrial oxidation of cytosolic NADH [65,66]. Remarkably, the activation of GOT2 by acetylation is opposite to generally inhibiting action of acetylation in mitochondria.

In contrast to acetylation, which is supposed to upregulate glycolysis, regulation by the negatively charged acylations is not well-characterized. Yet, sirtuin 5 knockout lowers glycolytic flux in mouse hepatocytes [24] and enhances susceptibility to kainate-induced seizures [31]. On the other hand, proteins carrying malonylation and succynilation sites are enriched in the glycolysis pathway [19,24], although glutarylation of proteins, including those of glycolysis, is the least characterized regulatory mechanism. Our work on multiple changes in glutarylation levels of proteins after seizures (Figure 1), and strong positive and negative correlations of these modifications in the brain proteins with the levels of glutaryl-CoA producer OADH and deglutarylase SIRT5 correspondingly (Figure 3), warrants further studies on the glutarylation system of the brain as a potential target in epilepsy treatment.

## 4. Materials and Methods

### 4.1. Reagents

When not specified otherwise, chemicals were obtained from Merck (Helicon, Moscow, Russia). PTZ (Merck, #P6500, Helicon, Moscow, Russia), thiamine·HCl (SERVA Electrophoresis GmbH, #36020, Helicon, Moscow, Russia), pyridoxal·HCl (PanReac AppliChem, #A0960, Helicon, Moscow, Russia) were dissolved at the day of injections and pH was adjusted to neutral with NaOH, if necessary. NAD^+^ was purchased from GERBU (#1013, Biolab-Ltd., Moscow, Russia). Formate dehydrogenase was obtained from Federal Research Center of Biotechnology/Innotech MSU (Moscow, Russia). Deionized MQ-grade water was used for solution preparations.

### 4.2. Animals

Wistar male rats were obtained from the Russian Federation State Research Center Institute of Biomedical Problems RAS (IBMP). The animals were kept in standard conditions of a 12 h light and 12 h dark day cycle, with free access to water and food. Further details of the animal husbandry and group assignment were as described earlier [26]. The adult rats of 11–12 weeks old (weighting 320 ± 30 g) were used for the single PTZ-induced seizure episode. Because the PTZ kindling required three weeks, the rats of 7–8 weeks old (weighting 140 ± 10 g) were used in the kindling model of epilepsy. Hence, by the day of analysis, the rats after a single and chronic seizures were of the same age and weight.

In total, 30 rats were randomly selected from the six animal groups, i.e., the control, single seizure and chronic seizures groups, each without and with vitamin administration. MS and associated analyses of the brain homogenates performed in this work was performed on 5 animals within each of the six groups.

### 4.3. Modelling Seizures by Single or Repeated Administration of Pentylenetetrazole

Intraperitoneal administration of PTZ in saline was employed. Induction of a single seizure episode by PTZ was described previously [26]. Briefly, up to three repeated PTZ injections, each of 25 mg/kg, were administered to achieve stages 4–5 of the Racine scale. After each PTZ injection, the severity of seizures of animals in individual cages (OpenScience, Moscow, Russia) was visually assessed for 15 min. Each visual assessment encompassed no more than 4 rats at a time (2 after the PTZ injection and 2 after the saline (0.9% NaCl) injection). Scores were registered every minute of the observation, and an epileptic seizure of a maximum score was noted for the given minute of observation.

Based on published data [29,30,67,68,69], we elaborated the kindling model of epilepsy as shown in Figure 6. As we aimed at comparison of the single PTZ injection and PTZ, kindling models, the younger rats were used for the PTZ kindling than in the single PTZ injection (see Section 4.2). Accordingly, the length of the repeated PTZ introductions in our kindling model (21 days) was close to the published results on the PTZ kindling in the young rats (20 days) [69]. When adult rats were taken into the study, longer times of the kindling was found to be optimal (25–30 days) [68], suggesting age-dependent effects. As a result, in our PTZ kindling model, intraperitoneal injections of PTZ at a dose of 37.5 mg/kg (sub-convulsive dose) were administered 3 times a week for 3 weeks, representing 9 injections or 37.5 mg of PTZ in total (Figure 6). Each of the injections was followed by the visual assessment of the seizures, as described above. The development of kindling was confirmed by PTZ injecting as in model of a single PTZ-administration a week after the last sub-convulsive dose injection. 24 h before and 15 min after the final PTZ injection, half of the control, single seizure and kindling animals received injections of either a mixture of vitamins B1 and B6 at a dose of 100 mg/kg each or equivalent volumes of saline. 24 h later the rats were sacrificed by decapitation using a guillotine (OpenScience, Russia). Euthanasia by decapitation is considered one of the least stressful methods to kill animals such as young rats [70,71,72]. After each use, the guillotine was washed thoroughly with water and ethanol, so that no blood smell could distress another animal. The procedure was performed according to recommended protocols [70,71,72]. Animal experiments and all the described procedures, including euthanasia by decapitation, were approved by Bioethics Committee of Lomonosov Moscow State University (protocols 69-o from 9 June 2016 for single PTZ-induced epilepsy and 139-a from 11 November 2021 for kindling model of epilepsy). The brains were excised and transferred onto ice, where the cerebral cortices were separated for freezing in liquid nitrogen 60–90 s after decapitation. The cortices were stored at −70 °C.

### 4.4. Homogenization and Extraction of Rat Tissues

Homogenization of the brain tissue and sonication of homogenates was carried out according to the previously published protocol [17], as well as methanol-acetic extraction [73].

### 4.5. Mass-Spectrometric Detection and Quantification of Acylations and Sirtuin 2 Level

The tissue samples were subjected to SDS-PAGE with a concentration of the separating gel of 10%. The polyacrylamide gel areas between the 20 kDa and 75 kDa, corresponding to most of the proteins of cerebral cortex homogenates were excised and subjected to in-gel digestion and Nano-LC-MS/MS analysis, as described earlier [17,21]. Mass spectra were acquired in positive MS mode, tuned for tryptic peptides. The spectra acquisition consisted of an orbitrap full MS scan (FTMS; resolution 60,000; *m*/*z* range 400–2000) followed by up to 15 LTQ MS/MS experiments (Linear Trap; minimum signal threshold: 500; dynamic exclusion time setting: 30 s; singly charged ions were excluded from selection, normalized collision energy: 35%; activation time: 10 ms). Raw data processing, protein identification and acylation assignment of the high resolution orbitrap data were performed by PEAKS Studio 8.0 (Bioinformatics Solutions, Waterloo, ON, USA). False discovery rate was set to <1%.

The peptides identified as described above were then filtered using a Python script according to the following criteria:Cleavage sites (the ends of the peptides) occur only after lysine and arginine or methylated lysine and arginine, which correspond to the cleavage sites of the trypsin used to obtain the peptides,Removal of starting methionine is allowed,The real charge of the peptide does not exceed that theoretically calculated from the sequence, assuming N-terminus, Lys, Arg and His to potentially be charged in the positive MS mode,The peptide is unique (i.e., occurs in only one protein according to the assignment of the peptides during their identification).

To assess the PTM abundancy separately from the alterations in the protein expression, the modified peptide was normalized to a highly abundant unmodified peptide from the same protein. Since not all of the peptides could be detected in all samples, the Skyline platform was used for manual quantification of such peptides [74], using the retention times identified by PEAKS Studio similarly to the previous papers [15,17,20,75]. The 43 acylation sites were assessed manually of the 99 sites identified in total (Appendix A).

To determine sirtuin 2 level, the peak area of its abundant peptide was normalized on the sum of peak areas of 9 peptides from actin and tubulin which passed the filtration criteria similarly to the previously described protocol [15].

### 4.6. Enzymatic Assays

The activity of 2-oxoglutarate dehydrogenase complex was measured as previously described [43]. The activity of the PDH complex was determined according to a published protocol for the enzyme assay in tissue homogenates [76]. The activities were assayed in 0.2 mL of the reaction medium spectrophotometrically in transparent 96-well microplates, using a Sunrise plate reader (Tecan, Grödig, Austria).

### 4.7. Western-Blotting Quantification of the Protein Levels of 2-Oxoadipate Dehydrogenase, Sirtuin 3, and Sirtuin 5

The levels of sirtuin 3, sirtuin 5 and OADH were estimated by western blotting using primary antibodies from Cell Signaling Technology (Danvers, MA, USA)—#8782 and #5490 for sirtuin 5 and sirtuin 3, respectively—and Thermo Fisher Scientific (Waltham, MA, USA)—#PA5-24208 for OADH. The primary antibodies for sirtuin 5, sirtuin 3 and OADH protein were used in 1:2000, 1:2000 and 1:400 dilutions, respectively, with the secondary anti-rabbit HRP-conjugated antibodies from Cell Signaling Technology, #7074 (Danvers, MA, USA). The relative quantification of chemiluminescence was performed in ChemiDoc Imager (Bio-Rad, Hercules, CA, USA) and Image Lab software version 6.0.1 (Bio-Rad, Hercules, CA, USA). Normalization of the protein levels to the total protein in the corresponding gel lanes was performed using the protein fluorescent quantification with 2,2,2-trichloroethanol, similarly to the published procedure [77]. The band intensities from different membranes were compared across all the membranes after the normalization to the levels of the common samples present in independent membranes.

### 4.8. NAD^+^ Quantification

NAD^+^ quantification in methanol-acetic extracts of rat cortices was performed according to the previously published protocol [78].

### 4.9. Statistical Analysis

All data were analyzed using RStudio R Statistical Software (version 4.1.1; R Foundation for Statistical Computing, Vienna, Austria). Changes in acylations, NAD^+^ level and protein expression of sirtuins and dehydrogenases of 2-oxo acids are presented as box-and-whisker plots, showing quantiles of each sample distribution. Outliers were determined and excluded separately for each group of PTZ administration factor using the interquartile range rule (1.5 IQR criterion). The data were tested for normality by the Shapiro–Wilk test. Null hypothesis was rejected in only 5% of groups in all measured parameters (Appendix A). Thus, two-way analysis of variance (ANOVA) and post hoc Tukey’s test were used for group comparisons, as ANOVA is quite tolerant of deviations from normality [79,80,81]. Spearman correlation coefficients were calculated and presented as heatmaps. The two-tailed *p* values 0.05 were considered to indicate statistically significant differences, shown in the figures.

## Figures and Tables

**Figure 1 ijms-23-12302-f001:**
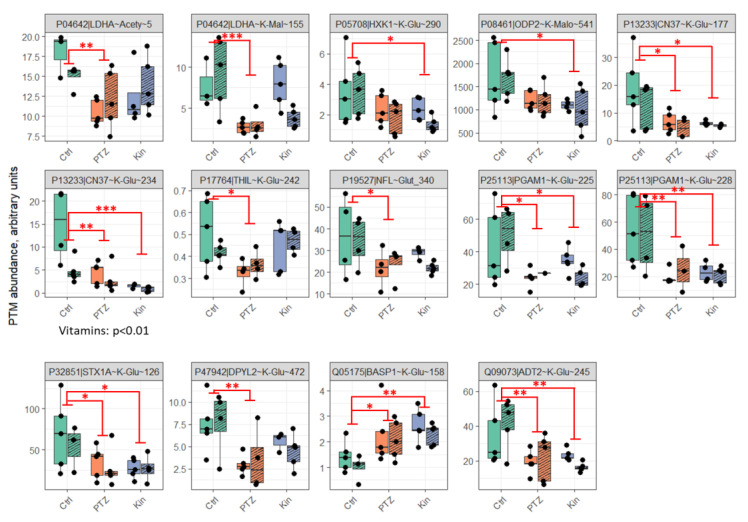
Acylations of the rat brain cortex proteins, found to change significantly after PTZ-induced seizures. Shading denotes administration of vitamins. In all the presented cases, the factor of PTZ treatment is significant (*p* ≤ 0.05) according to two-way ANOVA, while the factor of vitamins administration is not, except for P13233|CN37~K-Glu~234 (indicated below the graph). *, ** and *** signify *p* ≤ 0.05, *p* ≤ 0.01 and *p* ≤ 0.001 for comparisons of single PTZ-induced seizure (PTZ), seizure upon pharmacological kindling with PTZ (Kin) and the control animals (Ctrl) according to post hoc Tukey test. The level of acylation in each of the presented animal group is quantified as described in Methods. Proteins and their modified residues are defined in the grey sections of the graphs. The UNIPROT IDs of the proteins are shown, with the panel order of the graphs according to increasing number of the identifiers. Acety—acetylation, Mal—malonylation, Glu—glutarylation. Each of the six studied groups comprised five animals. However, some data points are missing on the plots (not more than 2 per group of 10 control or exposed-to-PTZ animals) or excluded as outliers (not more than 3 per group of 10 control or exposed-to-PTZ). All sample sizes are given in Appendix A.

**Figure 2 ijms-23-12302-f002:**
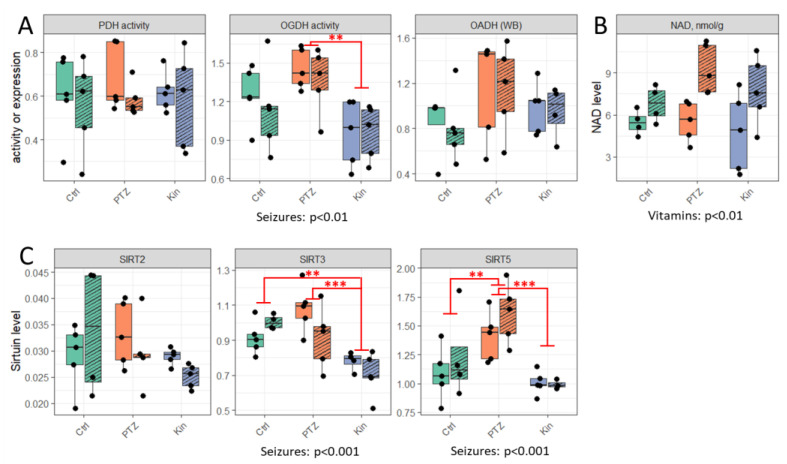
Effects of PTZ-induced seizures on acylation–deacylation regulators. (**A**) Activities of the brain pyruvate (PDH) and 2-oxoglutarate (OGDH) dehydrogenase complexes and protein expression of 2-oxoadipate dehydrogenase (OADH) measured by western blotting (WB). (**B**) NAD^+^ level in methanol-acetic extracts of the brain cortices. (**C**) Protein expression of brain sirtuin 2, 3 and 5 measured as described in Methods. ** and *** signify *p* ≤ 0.01 and *p* ≤ 0.001 according to post hoc Tukey test for the groups of the PTZ administration factor. Significant *p* values for the factors of PTZ treatment or vitamins administration are indicated below graphs. Each of the six studied groups comprised five animals. When fewer experimental points are shown on the plots, this is due to merged values and/or removed outliers (not more than 2 per group of 10 control or exposed to PTZ animals). All sample sizes are given in Appendix A.

**Figure 3 ijms-23-12302-f003:**
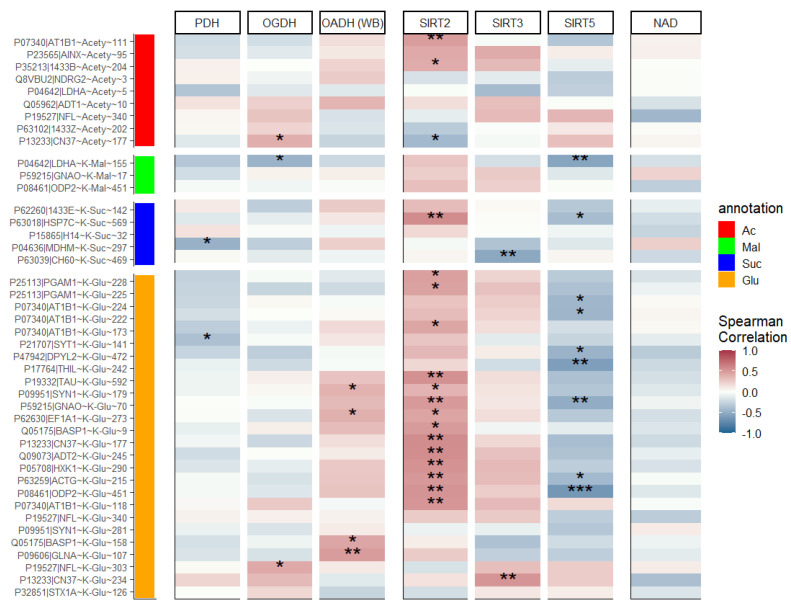
Spearman correlations of acylation levels of rat brain proteins with the activities of 2-oxoacid dehydrogenases PDH and OGDH, the protein expression of OADH measured by western blot (WB), the level of NAD^+^ and sirtuins 2, 3 and 5, measured as described in Methods. *, ** and *** signify *p* ≤ 0.05, *p* ≤ 0.01 and *p* ≤ 0.001 for Spearman rank correlations between each acylation site and biochemical parameter. Some correlation sample sizes deviate from the maximal number of studied animals (n = 30) because of missing data and/or outlier exclusion. All correlation sample sizes are given in Appendix A.

**Figure 4 ijms-23-12302-f004:**
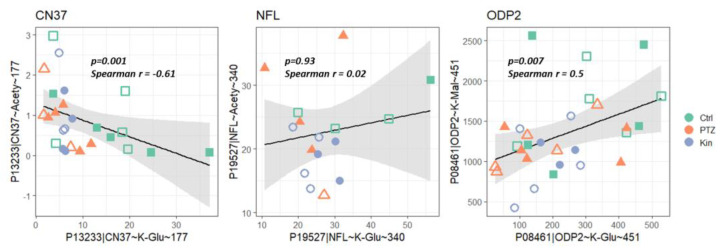
Spearman correlations between different types of acylation at a single site. Three sites modified by two different acyl residues in proteins indicated above the graphs, are shown. Hollow points correspond to animals with vitamin supplementation. Spearman correlation coefficients and corresponding *p* values are presented on the graphs. Sample sizes used for the correlations are provided in Appendix A.

**Figure 5 ijms-23-12302-f005:**
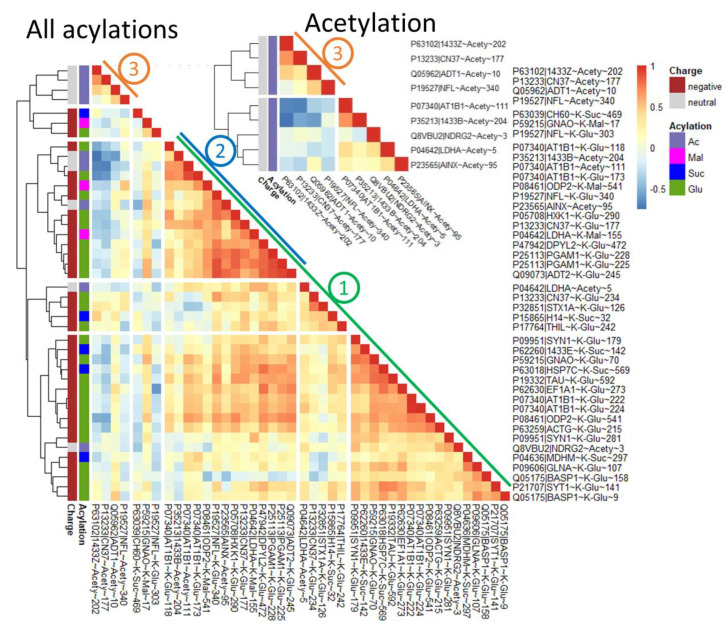
Spearman correlation matrix of 43 acylation sites plotted as a heatmap. Acylation sites are clustered by complete linkage clustering method. Specific clusters discussed in the text are numbered. All correlation sample sizes are given in Appendix A.

**Figure 6 ijms-23-12302-f006:**
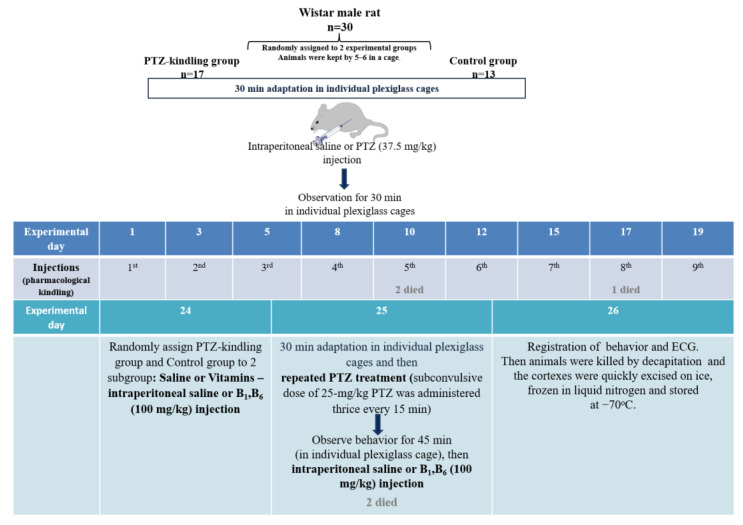
The flowchart of the kindling model of epilepsy. The pharmacological kindling was made during 9 days with the 37.5 mg/kg of PTZ. Three of 17 rats randomly assigned to the kindling group died. After a 5-day break, the rats were randomly assigned to the saline or B1 and B6 vitamins groups and the corresponding injections were made on days 24 and 25 as described in the scheme. Two more rats receiving PTZ died—five in total. That is, 13 of 30 rats did not receive PTZ, of which 6 rats were assigned to control group receiving saline and 7 rats received vitamins. Of 17 rats receiving PTZ 12 rats survived till the end. Of these rats 6 received vitamins and 6 received saline.

**Table 1 ijms-23-12302-t001:** Changes in enzymatical components of the brain acylation system in different epilepsy models; n.d., not determined.

Regulator	Brain Region	Model	Direction of Changes		Reference
			Published	This Work	
sirtuin 5	hippocampus	kainate, single	increase	increase	[31]
sirtuin 3	hippocampus	kainate, chronic	decrease	decrease	[32]
sirtuin 3	hippocampus	pilocarpine, single, in juvenile rats	decrease	n.d.	[33]
sirtuin 3	hippocampus	PTZ, chronic	decrease	decrease	[34]
PDHC	hippocampus	PTZ, chronic	decrease	n.d.	[34]
OGDHC	hippocampus	PTZ, chronic	decrease	n.d.	[34]
PDHC	hippocampus	pilocarpine, chronic	decrease	n.d.	[35]
OGDHC	hippocampus	pilocarpine, chronic	decrease	n.d.	[35]

## Data Availability

The data presented in this study are available in this article (summarized in figures and tables, including Appendix A). The raw data are available on request from the corresponding author.

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
