# Peer review of "The Brain Protein Acylation System Responds to Seizures in the Rat Model of PTZ-Induced Epilepsy"

_ijms, 2022, doi:10.3390/ijms232012302_

Round 1

Reviewer 1 Report

The authors show results indicating altered acylation status in metabolic enzymes caused by PTZ-induced seizures in a mouse model and suggest this might be a target for treatment in seizure disorders.  The manuscript is well written and their findings are well described however there seems to be a missing link in causality. The authors need to do a better job of showing that seizures in general cause the observed alteration of acylation status and that it is not caused by the administration of PTZ. In other words, are PTZ induced one-time epileptic events the same as a natural epileptic event?  As I understand it, the controls in the experiment are the same mice with saline injected instead of PTZ.  A stronger control might be a mouse that does not have seizures with PTZ injected which might not be possible with this model. The only evidence from another model the authors give is the observation that SIRT3 is decreased in mice with kainic acid induced seizures, which again, is not a natural seizure. This would be a much stronger paper if the authors could find similar data in a more natural model or if they can do a better job convincing the reader that PTZ-induced seizures adequately mimic seizure disorders in humans. 

There are also these minor concerns:

Please provide the n number for mice in the single-seizure group. I did not see it in the manuscript but I might have just overlooked it. 

Line 393: delete the extra "5" before the word "to"

Line 426: the word "rat" needs to be plural in the sentence "13 of 30 rat..."

Author Response

We sincerely thank the reviewer for the critical comments. We addressed all of them upon the revision. The point-by-point answers are provided below.

"The authors show results indicating altered acylation status in metabolic enzymes caused by PTZ-induced seizures in a mouse model and suggest this might be a target for treatment in seizure disorders.  The manuscript is well written and their findings are well described however there seems to be a missing link in causality. The authors need to do a better job of showing that seizures in general cause the observed alteration of acylation status and that it is not caused by the administration of PTZ. In other words, are PTZ induced one-time epileptic events the same as a natural epileptic event?  As I understand it, the controls in the experiment are the same mice with saline injected instead of PTZ.  A stronger control might be a mouse that does not have seizures with PTZ injected which might not be possible with this model. The only evidence from another model the authors give is the observation that SIRT3 is decreased in mice with kainic acid induced seizures, which again, is not a natural seizure. This would be a much stronger paper if the authors could find similar data in a more natural model or if they can do a better job convincing the reader that PTZ-induced seizures adequately mimic seizure disorders in humans. "

We use an established, state-of-the-art model of the epileptic seizures, and we clearly define this by indicating the type of seizures we study. Upon the revision, we checked and edited the text for this clarity once more. Of course, any animal model has its limitations, yet the models are used, also for the human drug tests. Most important, understanding the molecular mechanisms of the model greatly helps to deal with the limitations. Being an antagonist of GABAA receptor, the “non-natural” PTZ mimics a number of natural causes, from the low synaptic GABA levels till the GABAA receptor mutation, all leading to dysfunctional GABAA receptor. This specific binding of PTZ to GABAA receptor cannot change any other protein target but the GABAA receptor.

To address the reviewer remark, in the revised version, we mentioned these mechanistic aspects and extended the discussion of coincidence of findings in different models of epilepsy, all testifying to the observed changes in the brain acylation system being caused by seizures, not by a particular agent.  The control suggested by the reviewer can hardly be applied, as there is no fixed PTZ dose inducing the seizures to be compared to another fixed dose not inducing seizures. Besides, the dose dependence is inherent in any response – so, one always may object that at the non-seizure doze the PTZ effect is not achieved.

Our results on changed acylation of proteins involved in energy metabolism, agree with abnormal energy expenditure during seizures, thus further supporting the view that our findings relate to seizures rather than to PTZ itself.

"There are also these minor concerns:

Please provide the n number for mice in the single-seizure group. I did not see it in the manuscript but I might have just overlooked it."

  • We previously provided this in Methods (section on Mass-Spectrometric Detection). However, in view of the remark, we now transferred this to the section “Animal experiments” and also added to the figure legends. 30 rats in total were analyzed by MS, 5 animals in each of 6 PTZ&vitamins groups.

"Line 393: delete the extra "5" before the word "to""

  • Done

"Line 426: the word "rat" needs to be plural in the sentence "13 of 30 rat...""

  • Done

Reviewer 2 Report

In this study, the authors generated rat models for acute and chronic seizures by PTZ treatment and found changes in protein acylation, enzymatic activities of 2-oxo acid dehydrogenases, NAD+ level, and expressions of SIRTs in the brain. I felt that the manuscript dealt with an interesting topic. However, in my opinion, the authors need to clarify the following points, prior to publication.

1) At the beginning of the “Results,” the authors described the 1number of acylated proteins and sites without any explanation. It would make it difficult for the reader to understand what the experiment was performed. The authors should briefly describe for why they did the experiment, what model they created, what specimens they used, and what they measured, and so on. It would be better to show and explain the experimental schedule in Figure 6 first.

2) On the other hand, the “Results” is very redundant, describing not only the results of the experiment, but also a discussion by the authors. The author's discussion related to the results should be separated within the “Discussion” section. The results should be stated more concisely. 

3) In this study, acute and chronic epilepsy models were created by administering PTZ and further they were administrated with vitamins. Analysis of these rat groups has yielded a number of data on the acylation, but none of the data appear to be consistent. For example, the authors described, “our previous data on protective effect of the combined administration of vitamins B1 and B6 in the pentylenetetrazole (PTZ)-induced seizures (line 81).” However, in the present data, administration of vitamins did not alter the acylation level in the brain (Fig.1). Moreover, the acylation was decreased in both the acute and chronic PTZ models, but neither enzymatic activities of 2-oxo acid dehydrogenases, NAD levels, nor SIRTs expression varied in the two models. It is impossible to draw a consistent molecular pathogenesis associated with acylation from these data. The authors need to show evidence that acylation is indeed involved in the pathogenesis of epilepsy.

 4) This is in the same line of question. In this study, the authors detected changes in acylation in the brain after an epileptic seizure occurs, but there is no evidence that this is the cause of epileptic seizures. For example, is it possible that the change in acylation in the model is due to a pharmacological effect of PTZ itself? Are similar acylation changes observed in other epilepsy models? Furthermore, does administration of antiepileptic drugs suppress decreases in the protein acylation? Also, does administration of an inhibitor of SIRTs prevent the decrease in acylation, leading to improvement of seizures? The authors should show data linking the protein acylation to epileptic seizures.

5) This is also in the same line of question. The authors stated, “Thus, the abnormal energy expenditure in the PTZ-induced seizures is associated with alterations in the acylation level of the proteins involved in central energy metabolism (line 134).” However, there is no evidence that the protein acylation causes dysfunction of energy metabolism. Does the acylation changes on the proteins observed in the rat models actually alter their protein function? Did the authors actually check changes in respiratory chain function, glycolytic changes, and changes in ATP level in the models? The authors should show data linking the protein acylation to change of energy metabolism.

6) In Figure 3, the authors showed a strong positive correlation between SIRT2 and acylation including acetylation, succinylation and glutarylation. Why was the positive correlation observed since SIRT2 is a deacylation enzyme?

7) In Figure 5, The authors showed that acylation cites are clustered. Is this clustering of acylation a phenomenon specific to epilepsy? I am wondering that many positive correlations were observed simply because Lys residues on protein surfaces are generally easier to be acylated?

Author Response

Many thanks for the thorough analysis of our manuscript. We revised it according to the critical comments. Our answers are provided below.

"1) At the beginning of the “Results,” the authors described the 1number of acylated proteins and sites without any explanation. It would make it difficult for the reader to understand what the experiment was performed. The authors should briefly describe for why they did the experiment, what model they created, what specimens they used, and what they measured, and so on. It would be better to show and explain the experimental schedule in Figure 6 first."

  • We added the requested description of a model in a way that does not result in redundancy, as the model is described in Methods, and the journal format has Methods at the end of the manuscript. Our goal and aims (why we did, what model we used etc) are described/justified in Introduction. We hope we have currently addressed this remark sufficiently at the beginning of results, too.

"2) On the other hand, the “Results” is very redundant, describing not only the results of the experiment, but also a discussion by the authors. The author's discussion related to the results should be separated within the “Discussion” section. The results should be stated more concisely. "

  • We removed some discussion from results. Only specific comments and conclusions from the obtained data are left for better understanding of the paper and its logic.

"3) In this study, acute and chronic epilepsy models were created by administering PTZ and further they were administrated with vitamins. Analysis of these rat groups has yielded a number of data on the acylation, but none of the data appear to be consistent. For example, the authors described, “our previous data on protective effect of the combined administration of vitamins B1 and B6 in the pentylenetetrazole (PTZ)-induced seizures (line 81).” However, in the present data, administration of vitamins did not alter the acylation level in the brain (Fig.1). Moreover, the acylation was decreased in both the acute and chronic PTZ models, but neither enzymatic activities of 2-oxo acid dehydrogenases, NAD levels, nor SIRTs expression varied in the two models. It is impossible to draw a consistent molecular pathogenesis associated with acylation from these data. The authors need to show evidence that acylation is indeed involved in the pathogenesis of epilepsy."

           The previous data on the role of administration of B1 and B6 regarded totally different parameters than the current manuscript. In particular, the vitamins were shown to affect the levels and/or activities of the pyridoxal-dependent enzymes and related metabolites. These effects should not necessarily be reproduced in the acylation levels. Hence there is no data inconsistency in this regard. To clarify it further, we modified the text accordingly. In particular, we would like to draw the reviewer’s attention that also here we do show and discuss the effect of vitamins on NAD levels, even if these effects are not expressed in the acylation levels. Besides, the enzymatic activity of OGDH, Sirt 3 and Sirt 5 do change between the two models. This was and is shown in Figure 2 and discussed in the text. We have also shown, that among 43 quantified acylation sites, 13 decreased and 1 increased after seizures either single or chronic (or both), while others did not change significantly (Figure 1). In the text, we explain that the changes in the enzymes of the brain acylation system between the two models are brought about as a homeostatic response leading to no changes in the acylations between the two models. We added Table 1 to Discussion of coincidence of our and others results.

 "4) This is in the same line of question. In this study, the authors detected changes in acylation in the brain after an epileptic seizure occurs, but there is no evidence that this is the cause of epileptic seizures. For example, is it possible that the change in acylation in the model is due to a pharmacological effect of PTZ itself? Are similar acylation changes observed in other epilepsy models? Furthermore, does administration of antiepileptic drugs suppress decreases in the protein acylation? Also, does administration of an inhibitor of SIRTs prevent the decrease in acylation, leading to improvement of seizures? The authors should show data linking the protein acylation to epileptic seizures."

  • First of all, PTZ is an antagonist of GABAA. Specific binding of PTZ to GABAA receptor cannot change any other protein target but GABAA receptor. Hence, our observation of the changes in other proteins is due to the events induced by the PTZ binding, i.e. seizures, not the binding itself. This issue and consideration of other models is extended now in the Discussion, particularly Table 1.

Round 2

Reviewer 1 Report

I feel like the authors have sufficiently addressed my concerns.  This would still be stronger if it were in a "natural" seizure model but I understand that experiments of this type would be overly complex.  The authors adding details of the PTZ helps address this and gives the reader info that is needed. 

Author Response

Thank you for your review.

Reviewer 2 Report

I listed seven questions in the peer review process. However, the authors answered only four of them. The authors should answer all questions sincerely.

Author Response

We see that our answers to Reviewer 2 were not fully inserted. Probably there was a copy-paste failure or a cut-off in the system window. Sorry for this complication.

In this study, the authors generated rat models for acute and chronic seizures by PTZ treatment and found changes in protein acylation, enzymatic activities of 2-oxo acid dehydrogenases, NAD+ level, and expressions of SIRTs in the brain. I felt that the manuscript dealt with an interesting topic. However, in my opinion, the authors need to clarify the following points, prior to publication.
1) At the beginning of the “Results,” the authors described the 1number of acylated proteins and sites without any explanation. It would make it difficult for the reader to understand what the experiment was performed. The authors should briefly describe for why they did the experiment, what model they created, what specimens they used, and what they measured, and so on. It would be better to show and explain the experimental schedule in Figure 6 first.
-    We added the requested description of a model in a way that does not result in redundancy, as the model is described in Methods, and the journal format has Methods at the end of the manuscript. Our goal and aims (why we did, what model we used etc) are described/justified in Introduction. We hope we have currently addressed this remark sufficiently at the beginning of results, too.

2) On the other hand, the “Results” is very redundant, describing not only the results of the experiment, but also a discussion by the authors. The author's discussion related to the results should be separated within the “Discussion” section. The results should be stated more concisely.

 -    We removed some discussion from results. Only specific comments and conclusions from the obtained data are left for better understanding of the paper and its logic.

3) In this study, acute and chronic epilepsy models were created by administering PTZ and further they were administrated with vitamins. Analysis of these rat groups has yielded a number of data on the acylation, but none of the data appear to be consistent. For example, the authors described, “our previous data on protective effect of the combined administration of vitamins B1 and B6 in the pentylenetetrazole (PTZ)-induced seizures (line 81).” However, in the present data, administration of vitamins did not alter the acylation level in the brain (Fig.1). Moreover, the acylation was decreased in both the acute and chronic PTZ models, but neither enzymatic activities of 2-oxo acid dehydrogenases, NAD levels, nor SIRTs expression varied in the two models. It is impossible to draw a consistent molecular pathogenesis associated with acylation from these data. The authors need to show evidence that acylation is indeed involved in the pathogenesis of epilepsy.
-        The previous data on the role of administration of B1 and B6 regarded totally different parameters than the current manuscript. In particular, the vitamins were shown to affect the levels and/or activities of the pyridoxal-dependent enzymes and related metabolites. These effects should not necessarily be reproduced in the acylation levels. Hence there is no data inconsistency in this regard. To clarify it further, we modified the text accordingly. In particular, we would like to draw the reviewer’s attention that also here we do show and discuss the effect of vitamins on NAD levels, even if these effects are not expressed in the acylation levels. Besides, the enzymatic activity of OGDH, Sirt 3 and Sirt 5 do change between the two models. This was and is shown in Figure 2 and discussed in the text. We have also shown, that among 43 quantified acylation sites, 13 decreased and 1 increased after seizures either single or chronic (or both), while others did not change significantly (Figure 1). In the text, we explain that the changes in the enzymes of the brain acylation system between the two models are brought about as a homeostatic response leading to no changes in the acylations between the two models. We added Table 1 to Discussion of coincidence of our and others results.

 4) This is in the same line of question. In this study, the authors detected changes in acylation in the brain after an epileptic seizure occurs, but there is no evidence that this is the cause of epileptic seizures. For example, is it possible that the change in acylation in the model is due to a pharmacological effect of PTZ itself? Are similar acylation changes observed in other epilepsy models? Furthermore, does administration of antiepileptic drugs suppress decreases in the protein acylation? Also, does administration of an inhibitor of SIRTs prevent the decrease in acylation, leading to improvement of seizures? The authors should show data linking the protein acylation to epileptic seizures.
-    First of all, PTZ is an antagonist of GABAA receptor. Specific binding of PTZ to GABAA receptor cannot change any other protein target but GABAR. Hence, our observation of the changes in other proteins is due to the events induced by the PTZ binding, i.e. seizures, not the binding itself. This issue and consideration of other models is extended now in the Discussion, particularly Table 1.

5) This is also in the same line of question. The authors stated, “Thus, the abnormal energy expenditure in the PTZ-induced seizures is associated with alterations in the acylation level of the proteins involved in central energy metabolism (line 134).” However, there is no evidence that the protein acylation causes dysfunction of energy metabolism. Does the acylation changes on the proteins observed in the rat models actually alter their protein function? Did the authors actually check changes in respiratory chain function, glycolytic changes, and changes in ATP level in the models? The authors should show data linking the protein acylation to change of energy metabolism.
-       We do not claim a causative link between the observed phenomena. Our goal was to study the brain acylation system after seizures. To the best of our knowledge, our presented data on the acylome changes are unique. Based on published data, we conclude that the abnormal energy expenditure (known) is associated with changed acylation (shown). The association does not mean causality. In the discussion (section 3.4), we do provide published data to better assess the association.

6) In Figure 3, the authors showed a strong positive correlation between SIRT2 and acylation including acetylation, succinylation and glutarylation. Why was the positive correlation observed since SIRT2 is a deacylation enzyme?
-       Possible explanations for such phenomenon are considered in Results and Discussion. Firstly, negatively charged acylations may correlate positively with SIRT2 due to competition of such acylation with acetylation of the same site, which is exemplified by CN37 K177 acetylation (Figure 4). Secondly, the acetylation network comprises multiple components. Owing to this, the causality link “SIRT2 expression is high => acetylation is low due to high deacetylase activity” does not hold. The network is apparently organized in such a way that sirt2 expression is regulated according to the acetylation level. This is, e.g., observed in the positive correlation between the brain glutamate levels and OGDHC activity, even if glutamate is degraded through OGDHC (PMID: 29326069)

7) In Figure 5, The authors showed that acylation cites are clustered. Is this clustering of acylation a phenomenon specific to epilepsy? I am wondering that many positive correlations were observed simply because Lys residues on protein surfaces are generally easier to be acylated?
-        Most of the post-translational acylations occur at the exposed residues, not in the protein core. However, we do not perform the correlation analysis to assess the residue location/susceptibility. Correlations in Figure 5 indicate pairwise interplay between the system components when one of the components is changed. As all the studied animals are pooled in Figure 5, the correlations are observed for the changes that occur with seizures. The clusters of correlations may point to a common regulation of the parameters within specific clusters. Once again, the clusters relevance to epilepsy is defined by the fact that the correlations are built using the changes occurring after seizures. In supplementary figure S1 we compare the three studied networks separately, and the comparison is mentioned in the text.

Round 3

Reviewer 2 Report

The authors have satisfyingly answered all my questions.